# Selection on *BrFLC1* Is Related to Intraspecific Diversity of *Brassica rapa* Vegetables

**Jiahe Liu †, Xu Cai †, Yufang Li †, Yue Chen, Baozhen Gao, Runmao Lin, Jianli Liang, Xiaowu Wang** and **Jian Wu \***

Institute of Vegetables and Flowers, Chinese Academy of Agricultural Sciences, No.12, Haidian District, Beijing 100081, China; 13121270799@163.com (J.L.); caixu0518@163.com (X.C.); liyufang326815@163.com (Y.L.); caascy@163.com (Y.C.); rongruonuanxinfei@163.com (B.G.); linrunmao@caas.cn (R.L.); liangjianli@caas.cn (J.L.); wangxiaowu@caas.cn (X.W.)
* Correspondence: wujian@caas.cn
† These authors contributed equally to this work.

**Abstract:** Flowering time is important for *Brassica rapa* vegetables because premature bolting before harvest can lower yield and quality. *FLOWERING LOCUS C* (*FLC*) acts as a key repressor of flowering. In this study, we identified a nonsynonymous mutation at the 58th nucleotide of exon1 in *BrFLC1* (named as Pe1+58 (A/C)) by screening resequencing data of 199 *B. rapa* accessions and verified this mutation as being related to flowering time variation. Strong linkage inheritance was detected between this locus and a previously reported splicing site mutation at intron 6 of *BrFLC1* (Pi6+1 (G/A)), showing as co-occurrence of BrFLC1Pe1+58(A) and BrFLC1Pi6+1(G), named as haplotype H1: AG, or co-occurrence of BrFLC1Pe1+58(C) and BrFLC1Pi6+1(A), named as haplotype H2: CA. The frequency distribution of *BrFLC1* haplotypes skewed to the haplotype H1 in turnip, broccoletto, mizuna, komatsuna, and taicai, while it was skewed to the haplotype H2 in caixin, pak choi, zicaitai, and wutacai. The frequencies of the two haplotypes were comparable in Chinese cabbage. This indicated that *BrFLC1* haplotypes were related to *B. rapa* intraspecific diversification. Further analysis of a Chinese cabbage collection revealed that accessions from the spring ecotype preferred to keep H1: AG and almost all accessions from the summer ecotype were H2: CA. The early flowering haplotype of *BrFLC1* was purified in summer Chinese cabbage, indicating that BrFLC1 had been strongly selected during genetic improvement of summer Chinese cabbages. A significant difference in flowering time of $F_2$ individuals with the homologous BrFLC1Pi6+1(G) allele but different BrFLC1Pe1+58 (A/C) alleles, indicated that this locus had independent genetic effects on flowering time. The newly identified allelic diversity of *BrFLC1* can be used for breeding of resistance to premature bolting in *B. rapa* vegetables.

**Keywords:** *Brassica rapa*; *FLOWERING LOCUS C*; haplotype; flowering time; intraspecific diversification

## 1. Introduction

*Brassica rapa* comprises vegetables, fodders, and oilseed crops. *B. rapa* responds to vernalization as early as at the stage of germinated seed. This characteristic is not favorable for producing leafy *B. rapa* vegetables because premature bolting often happens due to low temperatures during early growth and leads to loss of commercial value. Breeding of *B. rapa* varieties with strict requirements on vernalization conditions, such as longer period and lower temperature, could be an efficient way to avoid premature bolting in the cultivation of *B. rapa* vegetables.

Plants have evolved a complex genetic network to ensure flowering and seed set during favorable environmental conditions [1]. *FLOWERING LOCUS C* (*FLC*) is one of the key genes involved in controlling the conditions for vernalization. *FLC* encodes a MADS-box transcription factor that inhibits flowering by directly binding to floral promoting genes such as *FLOWERING LOCUS T* (*FT*), *SUPPRESSPR OF COSTANS 1* (*SOC1*), and

*SQUAMOSA PROMOTER-BINDNG PROTEIN-LIKE 15* (*SPL15*) to block their transcription [2] In *Arabidopsis thaliana* most of the variation in flowering time is controlled by *FLC* and variation of the *FRI* allele that activates the expression of *FLC* [3,4].

The ancestor of *Brassica* species underwent a genome triplication event after divergence from *A. thaliana* [5]. Based on syntenic analysis between *B. rapa* and *A. thaliana*, three copies of *FLC* (*BrFLC1*, *BrFLC2*, and *BrFLC3*) were identified in *B. rapa* collinear regions [5,6]. The three syntenic copies were confirmed being functional in the regulation of flowering time by transferring the genes into *A. thaliana* [7]. Moreover, genetic analysis using different germplasm materials and segregating populations further confirmed that *BrFLC1*, *BrFLC2*, and the copy located in the non-collinear region, *BrFLC5*, are flowering repressors [8–13]. These studies revealed a few naturally occurred sequence variations of *FLC* orthologous genes in *Brassica* that affected their functions in flowering repression. SNPs located at splicing sites and resulting in alternative splicing patterns were reported for two *B. rapa FLC* homologues, *BrFLC1 and BrFLC5* [10,13]. Okazaki et al. identified a single-base deletion in exon 4 of *BoFLC2* that produced a non-functional allele in *B. oleracea* [14]. Wu et al. detected a 57-bp deletion across exon 4 to intron 4 that resulted in loss-of-function of *BrFLC2* [9]. Large InDels have often been identified in introns of *FLC* genes. In *B. napus* a 2.833 kb fragment insertion in the first intron of *BnFLC.A2* generates a loss-of-function allele that can promote flowering when this allele is introgressed into chromosome C2 [15] In contrast, Kitamoto et al. reported a large insertion of 5037 bp near the 5′ end of the first intron of *BrFLC2* related to the late flowering in *B. rapa* [12]; however, there was no apparent correlation between a 5678 bp insertion in the first intron of *BrFLC3* and bolting time. These function-related sequence variations could potentially be used in marker-assisted selection (MAS) for breeding of premature bolting resistant *B. rapa* vegetables.

Flowering time is an important environmental adaption and thus has been under selection during crop domestication. The domestication of *B. rapa* vegetables has been analyzed regarding leafy head formation in Chinese cabbage and tuber formation in turnips [16]. It is interesting that genes related to tuber formation in turnips were found to be involved in the flowering pathway [17] Flowering time was analyzed in different Chinese cabbage ecotypes to dissect the effect of modern breeding selection in ecotype improvement [18] In Chinese cabbage, incorporation of the elite alleles *BrFLC1* and *BrVIN3.1* is the determining genetic factor of the spring ecotype [18] However, it is still unclear whether *BrFLC* genes were selected during intraspecific diversification of *B. rapa*.

Here, we report a nonsynonymous mutation in exon 1 of *BrFLC1*, named as BrFLC1Pe1+58 (A/C), related to flowering time variation observed in three *B. rapa* germplasm collections containing 199, 236, and 829 accessions. Linkage inheritance was found between BrFLC1Pe1+58 (A/C) and a previously identified splicing site mutation, BrFLC1Pi6+1 (G/A). Further analysis revealed that BrFLC1Pe1+58 (A/C) had an independent effect on flowering time in an $F_2$ population segregating only at the BrFLC1Pe1+58 (A/C) locus. The frequency distribution of haplotypes comprising BrFLC1Pe1+58 (A/C) and BrFLC1Pi6+1 (G/A) loci in different subspecies varied according to their flowering pattern; late-flowering had a high frequency of the H1 (AG), while early-flowering had a high frequency of H2 (CA). We demonstrate that there was strong selection on *BrFLC1* haplotype H2:CA during summer Chinese cabbage improvement.

## 2. Materials and Methods

### 2.1. Sequence Variation in Genome Region of BrFLC1

To identify naturally occurring mutations in *BrFLC1*, we extracted the genomic sequence of the *BrFLC1* genic region (including 2 kb upstream and downstream of the gene body) from the Chiifu v3.0 Assembly (http://brassicadb.cn (accessed on 1 August 2021)) and then mapped all resequencing data to the reference sequence. Reseuqencing data was collected from a total of 391 *B. rapa* accessions. Among them, 199 accessions were from our previous work [16]; 192 were collected from a previously reported study [18]. First, raw reads were filtered using fastp (version 0.12.3) [19] with parameters '-z 4-q 20-u 30-n 5'.

Then, all of the clean reads were mapped to the extracted sequence using BWA-MEM (version 0.7.5a-r405) [20] with the default parameters. Finally, variants were called using SAMtools (version 0.1.19-44428cd) [21].

### 2.2. Plant Materials and Growth Conditions

A total of 236 *B. rapa* accessions from 11 cultivar groups (Supplementary Table S4) were used to investigate flowering time. The 236 *B. rapa* accessions were sown in pots in a greenhouse on 19 February 2016. The temperature varied from 15 to 28 °C. After 21 days of growth, these seedlings were transplanted into an open field in the Shunyi District of Beijing, China, in a randomized complete block design. Each of these accessions was grown in five replicates.

A large-scale *B. rapa* germplasm collection with 785 accessions, including 212 from the previous 236 accessions, was used to analyze the *BrFLC1* haplotype distribution frequency among *B. rapa* cultivar groups. The 785 accessions were sown in plug trays in a greenhouse on 16 August 2017. The temperature was 24–31 °C. Twenty-two days later, leaf samples were harvested for DNA isolation.

A segregating population was constructed from an accession, that had a heterozygous allele of BrFLC1Pe1+58 (AC) and a homologous BrFLC1Pi6+1 (G) allele. In total, 166 $F_2$ lines were used for analyzing flowering phenotype differences between the individuals with different alleles of BrFLC1Pe1+58 (A/C). Germinated seeds were sown into plug trays in a greenhouse in the Haidian District of Beijing, China on 22 January 2021. After 18 days, the seedlings were transferred into pots and grown in the greenhouse without climate control until they flowered. The photoperiod was set to a 14 h:10 h day: night rhythm with supplementary lighting by high pressure sodium lamp.

### 2.3. Evaluation of Bolting Time and Flowering Time

Bolting time and flowering time were evaluated for the germplasm collection using 236 accessions and the $F_2$ population segregating at the BrFLC1Pe1+58 (A/C) allele. Bolting time was recorded as the number of days from sowing to the first flower bud appeared (days to bolting, DTB). Flowering time was recorded as the number of days from sowing to the first flower opening. The survey ended at 150 days after sowing for the germplasm collection. Bolting time and flowering time were recorded as 150 DTB and 160 DTF, respectively, for the accessions that did not bolt or flower until this time point. The survey ended at 120 days after sowing for the $F_2$ population. Bolting time and flowering time were recorded as 120 DTB and 140 DTF, respectively, for the individuals that did not bolt or flower before this time point.

### 2.4. KASP Assay for BrFLC1 Genotyping

Genomic DNA was extracted from fresh young leaves using a modified CTAB protocol [22]. The DNA concentration was measured by NanoDrop (ND-1000, Thermor Fisher Scientific) and normalized to a concentration of 15 ng/μL. The allele-specific primers were designed carrying the standard FAM (5′-GAAGGTGACCAAGTTCATGCT-3′) and VIC (5′-GAAGGTCGGAGTCAACG-GATT-3′) tails and with the targeted SNP at the 3′ end. The forward primers BrFLC1Pe1+58_F1 and BrFLC1Pe1+58_F2 for the A and C alleles, and the common reverse primer BrFLC1Pe1+58_R were designed to produce a PCR product with a length of 48 bp (for the A allele) or 50 bp (for the C allele) for the KASP assay. The primers used to discriminate the alleles at the locus BrFLC1Pi6+1 were the same as those reported by Xi et al. [13]. The KASP assay procedure followed Xi et al. [13].

### 2.5. Statistical Analysis

Analysis of variance (ANOVA) by independent samples *t*-tests was performed in the SPSS version 22.0 statistical package (SPSS Inc., Chicago, IL, USA). One-way ANOVA was performed with flowering time as the test variable and genotype as the grouping variable.

## 3. Results

### 3.1. A Nonsynonymous Mutation Pe1+58(A/C) of BrFLC1 Differently Appeared among B. rapa Subspecies

A nonsynonymous mutation at the nucleotide 58 in the first exon of *BrFLC1*, named as BrFLC1Pe1+58 (A/C), was identified from analyzing resequencing data of a natural population comprising 199 diverse *B. rapa* accessions. The population included accessions from groups of turnip, Chinese cabbage, pak choi, wutacai, caixin, and zicaitai, and it had been used to represent the variation landscape of *B. rapa* in our previous study [16]. A total of 49 SNPs and six InDels in the genic regions of *BrFLC1* (including 2 kb upstream and downstream of the gene body) were identified (Supplementary Table S1). Among these mutations, only two SNPs occurred in the CDS regions, one of which resulted in a non-synonymous mutation (ACC → cCC) and the other that led to a synonymous mutation. This non-synonymous mutation locus was named as BrFLC1Pe1+58 (A/C). Furthermore, the genotype frequency of BrFLC1Pe1+58 (A/C) in the population with 199 diverse accessions was analyzed (Supplementary Table S1). There was a clear trend of genotype frequency in the subspecies with different flowering patterns. Among the 54 turnip accessions, 92.59% were of the A genotype. In contrast, none of the 30 caixin accessions and 13 zicaitai accessions were of the A genotype (Figure 1). This result suggested that the nonsynonymous mutation at BrFLC1Pe1+58 (A/C) was related to flowering time variation, since the turnip accessions were later flowering, while accessions from caixin and zicaitai were early flowering.

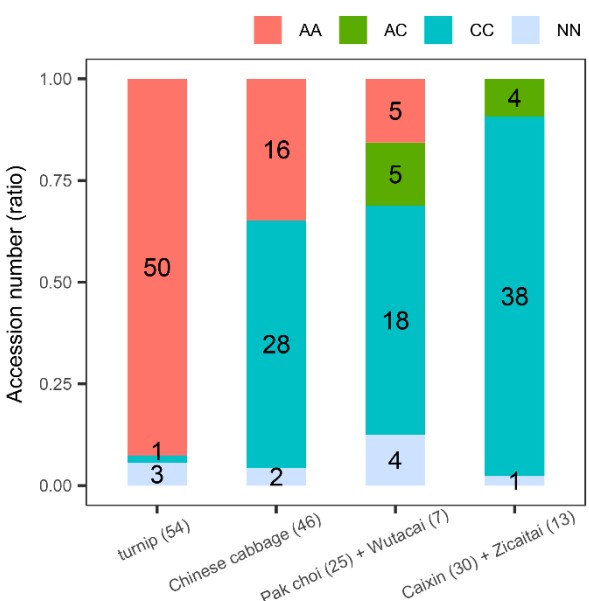

**Figure 1.** Frequency distribution of Pe1+58(A/C) genotypes in different *B. rapa* cultivar groups.

### 3.2. Pe1+58(A/C) of BrFLC1 Is Related to Flowering Time Variations among a Germplasm Collection of 236 B. Rapa

To determine whether BrFLC1Pe1+58 (A/C) was related to variation in flowering time, a germplasm collection with 236 *B. rapa* accessions from 11 subspecies or varieties was screened for allelic variations using a KASP assay. Among the 236 accessions, 123 carried the A allele; 100 carried the C allele; and 13 were heterozygous at this locus (Supplementary Table S2). Flowering time was assessed for the 236 accessions in an open field in the spring of 2016. The bolting time varied between 32 DTB to not bolting until the end of investigation (recorded as 150 DTB), while the flowering time varied between 45 DTF to not flowering at the end of investigation (recorded as 160 DTF). The DTB of plants homozygous for the C allele was mainly distributed between 60 and 100 days, while this was clustered at 80–150 days for plants homozygous for the A allele (Figure 2a). The accessions with a mutated C allele bolted significantly earlier than those with the wild type A allele ($p < 0.001$,

Figure 2b, Supplementary Table S2). This result indicated that the Pe1+58(A/C) allelic variation of *BrFLC1* was related to the flowering time variation in *B. rapa*.

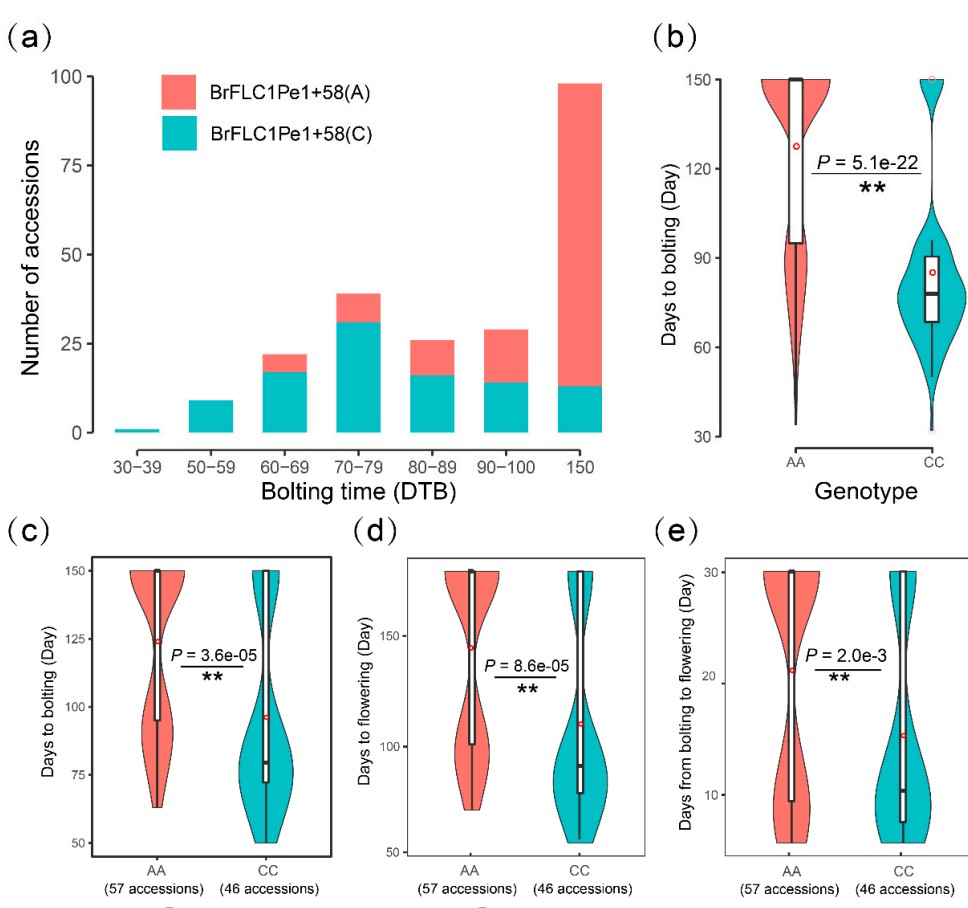

**Figure 2.** Frequency distribution of *BrFLC1* Pe1+58 alleles and bolting time distribution in a *B. rapa* germplasm collection (*n* = 236). (**a**) Distribution of BrFLC1Pe1+58 alleles with their associated bolting time. The heterozygous genotype was not included in the analysis because the number of accessions was too few; (**b**) distribution of bolting time for the accessions with different *BrFLC1* Pe1+58 alleles; bolting time (**c**), flowering time (**d**), and the time from bolting to flowering (**e**) according to their *BrFLC1* Pe1+58 alleles in the group of Chinese cabbage. Comparisons are done by Student-Newman-Keuls test with $\alpha$ = 0.01, and the marker (**) indicates the *p*-value <= 0.01.

Among the 236 accessions, 112 were Chinese cabbages (ssp. *pekinensis*). In contrast to the distribution bias for the BrFLC1Pe1+58 (A/C) allele in the groups of turnip, caixin, or pak choi, the Chinese cabbage had comparable frequencies of the two alleles. The frequencies in the Chinese cabbage group were 50%, 42%, and 8% for AA, CC, and AC plants, respectively. The DTB and DTF of Chinese cabbages with AA allele were significantly longer than those of accessions carried CC allele (Figure 2c,d). The accessions with AA alleles also took significantly longer from bolting to flowering (Figure 2e). This result further confirmed that this natural mutation is related to variation in flowering time.

*3.3. BrFLC1 Haplotype Consisting of BrFLC1Pe1+58 (A/C) and BrFLC1Pi6+1(G/A) Was Associated with the Intraspecific Diversification and Ecotype Diversification in B. rapa*

A linkage inheritance between BrFLC1Pe1+58 and a previously reported splicing site mutation in intron 6 [10] BrFLC1Pi6+1 (G/A) was detected (Figure 3a) from analysis of *BrFLC1* sequence variations. The linkage inheritance manifested as the co-occurrence of genotypes BrFLC1Pe1+58(A) and BrFLC1Pi6+1(G) or co-occurrence of genotypes Br-FLC1Pe1+58(C) and BrFLC1Pi6+1(A), and the linkage inheritance was strongest in the

turnip and caixin (Figure 3a). Based on the strong linkage inheritance of the two loci, two haplotypes of the combinations of BrFLC1Pe1+58 and BrFLC1Pi6+1 were proposed as H1: AG and H2: CA (Figure 3c). All of the turnip accessions showed haplotype 1 (H1: AG), and all of caixin and zicaitai accessions showed haplotype 2 (H2: CA) (Figure 3a,d). However, the Chinese cabbage group showed a mixture of two haplotypes.

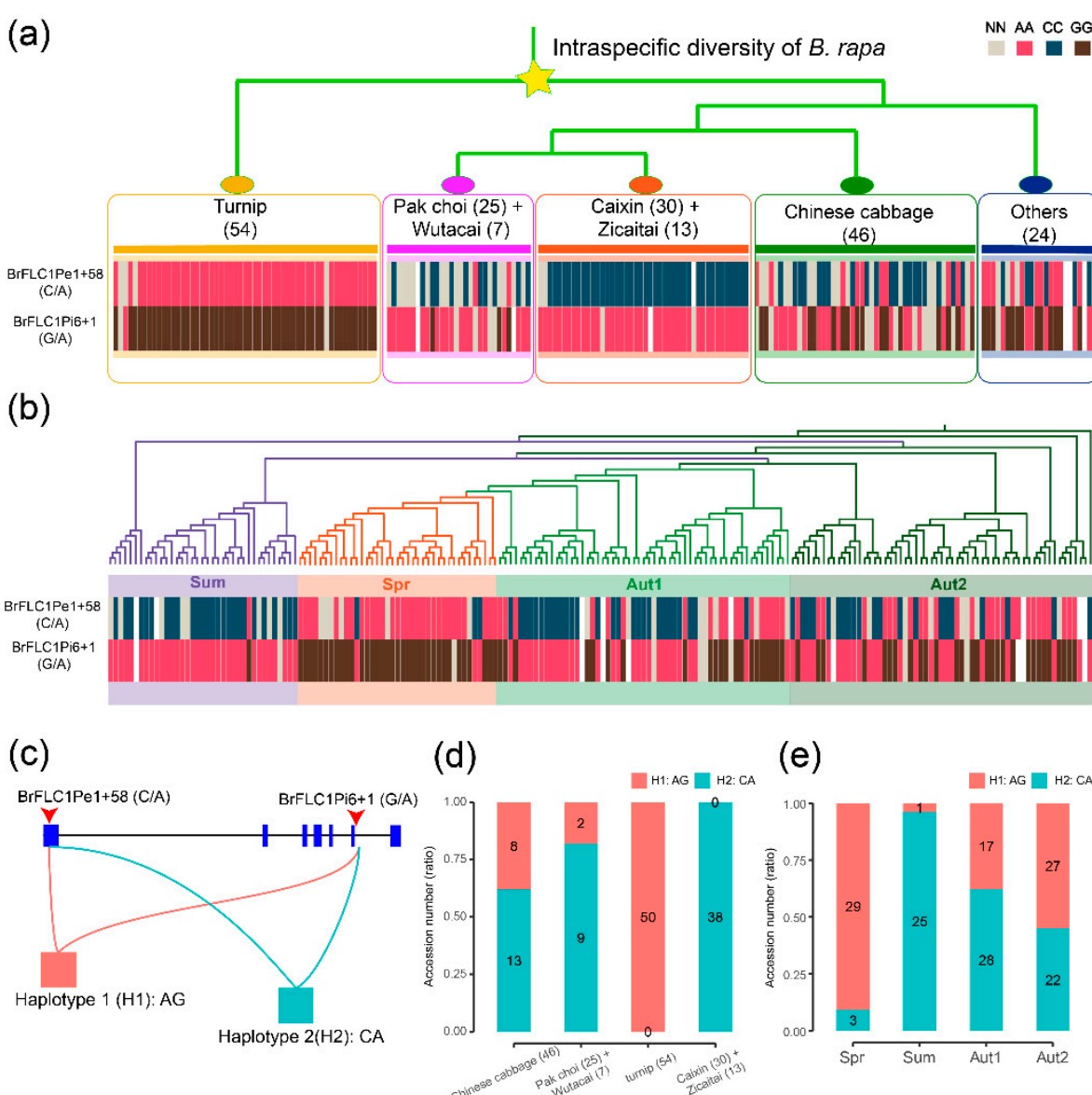

**Figure 3.** Distribution of haplotypes comprising BrFLC1Pe1+58 (A/C) and BrFLC1Pi6+1 (G/A) in different *B. rapa* subspecies. (**a**) Genotype distribution of BrFLC1Pe1+58 (A/C) and BrFLC1Pi6+1 (G/A) in a germplasm collection of 199 accessions [16]; (**b**) Genotype distribution of BrFLC1Pe1+58 (A/C) and BrFLC1Pi6+1 (G/A) in a collection of 192 Chinese cabbage accessions [18]; (**c**) Schematic model of haplotypes BrFLC1Pe1+58 (A/C) and BrFLC1Pi6+1 (G/A); (**d**) haplotype distribution in 120 of 199 *B. rapa* germplasm accessions; (**e**) haplotype distribution in 152 of 192 Chinese cabbage accessions. In (**d,e**), the accessions carrying heterozygous alleles in either BrFLC1Pe1+58 (A/C) or BrFLC1Pi6+1 (G/A) were not included.

To analyze the *BrFLC1* haplotype-phenotype association in Chinese cabbages, different ecotypes of Chinese cabbage were analyzed for the two haplotypes. The 192 Chinese cabbage accessions were divided into four ecotypes according to their growing seasons, including spring, summer, autumn 1, and autumn 2 [18]. In the present study, a phylogenetic tree was constructed based on the genome-wide SNPs (Figure 3b). The tree was in line with the ecotype division by growing season. Strong linkage inheritance and distribution

bias were also clear in Chinese cabbage. Around 90.63% of the accessions from spring Chinese cabbage showed haplotype 1 (H1: AG), and 96.15% of accessions from summer Chinese cabbage showed haplotype 2 (H2: CA) (Figure 3b,e). This result indicated that the strong linkage inheritance of the two loci was tightly associated with the diversification of different Chinese cabbage ecotypes.

To further explore the relationship between *BrFLC1* haplotypes and intraspecific diversity of *B. rapa,* an expanded germplasm collection with 829 diverse accessions was screened for genotypes of these two loci. These 829 accessions belonged to 11 subspecies or varieties, covering *B. rapa* crops as much as possible (Supplementary Table S3). Accessions of Chinese cabbage (252), pak choi (162), turnip (152), and caixin (62) occupied about 76% of this collection, because these varieties are rich in *B. rapa* germplasm. The genotyping results showed that among the 829 accessions, 390 were haplotype 1 (H1: AG) and 395 were haplotype 2 (H2: CA); 42 were heterozygous for both loci (Supplementary Table S3). The distribution of haplotype frequency differed among *B. rapa* cultivar groups (Figure 4). Two types could be observed. Type 1 was represented by turnip along with mizuna, komatsuna, taicai, and broccoletto, with a high proportion of haplotype 1. Type 2 was represented by pak choi together with caixin, wutacai, and zicaitai, with a high ratio of haplotype 2. The results did suggest that the *BrFLC1* haplotypes consisting of these two loci were associated with intraspecific diversification and have been under strong selection during *B. rapa* domestication. In terms of Chinese cabbage, as shown by the 192 accessions, *BrFLC1* haplotypes were strongly selected in the recent breeding procedure.

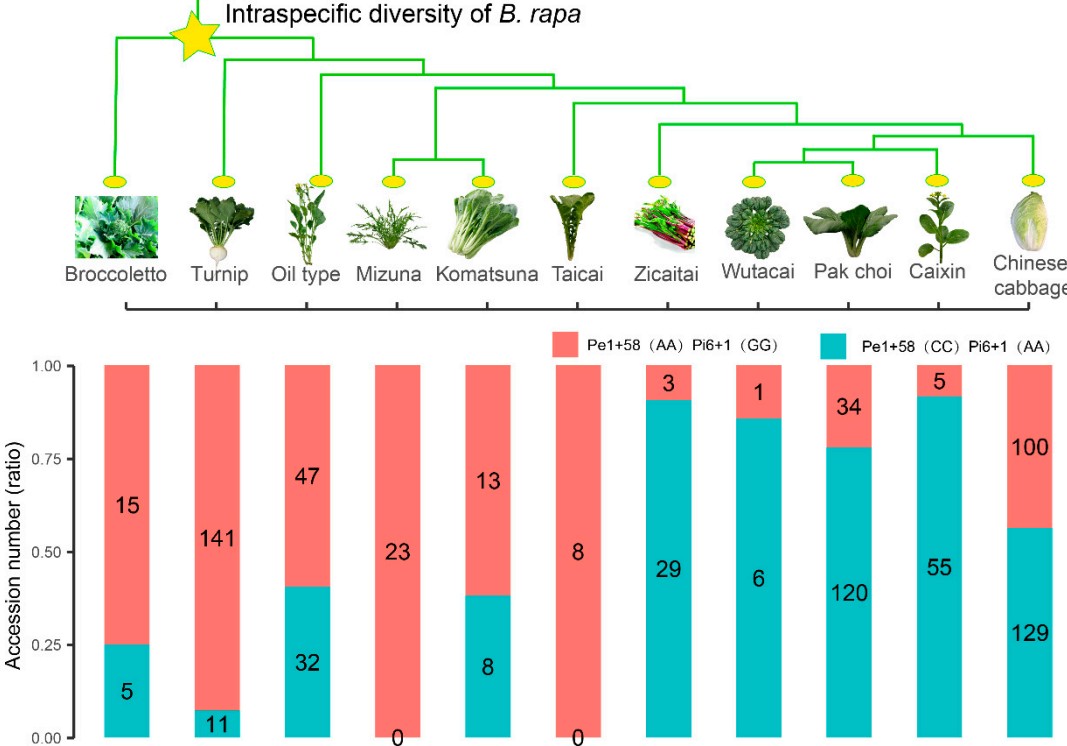

**Figure 4.** The frequencies of *BrFLC1* haplotypes in a germplasm collection with 829 *B. rapa* accessions. The phylogenetic tree was constructed based on the neighbor joining tree constructed with resequencing data of 524 accessions [23].

### 3.4. The BrFLC1Pe1+58 (A/C) Locus Poses Independent Genetic Effects on Flowering Time in B. rapa

The locus BrFLC1Pe1+58 (A/C) was revealed to have an independent genetic effect on flowering by analyzing an $F_2$ population segregating for this locus. From screening of 785 germplasm accessions, an oil-type accession HNAJHCZ was identified carrying heterozygous alleles BrFLC1P1e+58 (A/C) and homozygous on BrFLC1Pi6+1 (GG). The plant

was self-pollinated to generate a population segregating at the BrFLC1e+58 (A/C) locus to evaluate the genetic effect on variation in flowering time. In total, 166 $F_2$ individuals were investigated for bolting time and flowering time and were genotyped for BrFLC1P1e+58 (A/C) using the KASP assay. Among the 166 individuals, 39 carried the A allele; 37 carried the C allele, and 90 were heterozygous (Supplementary Table S4). The genotype ratio of A/H/C was 1:2.3:1, which was in line with the segregation ratio of the $F_2$ population. HNAJHCZ plants bolted at 63 DTB and flowered at 76 DTB. Among the $F_2$ individuals, the average bolting and flowering times of individuals with the A allele (62.0 DTB and 78.9 DTF) were later than those of CC individuals (55.4 DTB, 69.0 DTF) ($p < 0.05$ for DTB, $p < 0.05$ for DTF). However, there was no significant difference between individuals with AA and AC (58.1 DTB and 72.9 DTF), or between individuals with CC and AC genotypes. It is interesting that when considering flowering time as the period from the first bud appearing to the first flower blooming (DTF-DTB), the difference between AA (16.9 days) and CC (13.2 days) or between AA and AC (14.8 days) were both significant (Figure 5c). These results indicated that the nonsynonymous mutation BrFLC1Pe1+58 (A/C) affected *BrFLC1* function in repressing flowering in *B. rapa*. Although there was strong genetic linkage between the BrFLC1Pe1+58 (A/C) and BrFLC1Pi6+1 (G/A) loci, BrFLC1Pe1+58 (A/C) had an independent genetic effect and could contribute to the breeding of late bolting *B. rapa* vegetables.

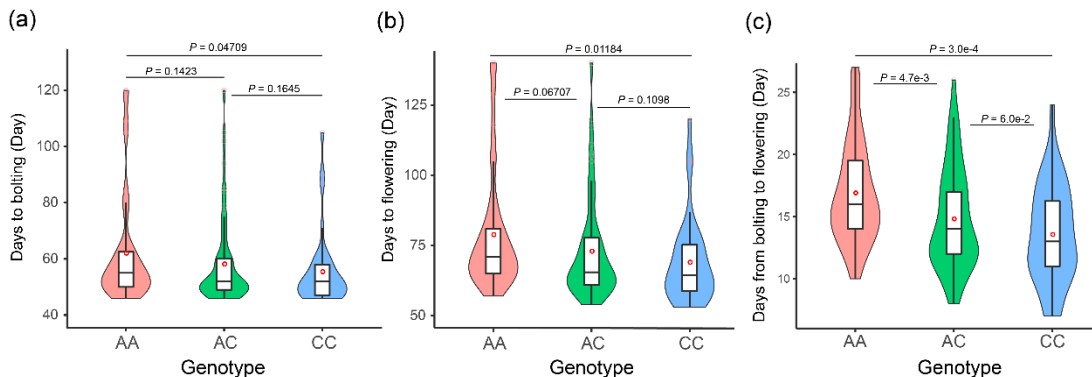

**Figure 5.** Distribution of bolting time (**a**), flowering time (**b**), and time from bolting to flowering (**c**) in an $F_2$ population. AA, homozygous A allele of BrFLC1Pe1+58; CC, homozygous C allele of BrFLC1Pe1+58; AC, heterozygous for BrFLC1Pe1+58. Comparisons are done by Student-Newman-Keuls test with $\alpha = 0.01$.

## 4. Discussion

In this study, a naturally occurring single nucleotide mutation, BrFLC1Pe1+58 (A/C), was detected in *BrFLC1*, and this mutation led to an amino acid shift from Thr to Pro. This mutation was significantly associated with flowering time variation in *B. rapa*.

The newly detected mutation BrFLC1Pe1+58 (A/C) was found to be in inherited linkage with a previously reported splicing site mutation, BrFLC1Pi6+1 (G/A) [10]. Mutated BrFLC1Pi6+1 (A) allele led to alternative splicing patterns and early flowering. To test whether the BrFLC1Pe1+58 (A/C) locus had an independent effect on flowering, a population segregating at this locus, but homologous to BrFLC1Pi6+1(G) was developed. By analyzing differences between flowering time of $F_2$ individuals with different BrFLC1Pe1+58 genotypes, we found that this locus affected flowering time independent of the BrFLC1Pi6+1 (G/A) locus. This result gives a hint for breeding of *B. rapa* crops that both loci need to be considered for enhancing premature bolting resistance.

Flowering time is one of the major traits under strong selection during crop domestication [24,25]. Previous research indicated that selection for flowering-related genes, especially *FLC* was a powerful force during the domestication of *Brassica* crops. *FLC*, encodes a core regulator in *B. napus* ecotypes improvement [26]. By pan-genome and large-scale re-sequencing of *B. rapa*, phylogenetic relationships have been established [23]. Therefore, we could analyze the relationship between the distribution of the *BrFLC1* haplo-

type in different *B. rapa* subspecies and intraspecific diversification. The haplotypes from combining these two loci were revealed to be associated with *B. rapa* diversity by analyzing a total of 829 germplasm accessions. Based on the previously constructed phylogenetic tree [23], different *B. rapa* subspecies were divided into two groups according to the position of taicai (Figure 4). The more ancient subspecies were skewed to carrying the haplotype of late flowering; however, the subspecies or varieties differentiated in China after the turnip was introduced from the Mediterranean–Middle Eastern area were biased to carrying the early flowering haplotype. This result was in line with the proposition that China is the secondary origin and differentiation center of *B. rapa* [27] Pak choi, wutacai, as well as caixin and zicaitai are prevalently cultivated in southern regions of the Yangzi River basin. Caixin flowers extremely early because its commercial organ is the flower stem. Our results indicate that selection, including natural and artificial, has played a significant role in *B. rapa* diversification.

In the group of Chinese cabbage, the case was more complicated. Su et al. divided 194 Chinese cabbage accessions into four ecotypes, spring, summer, autumn 1 and autumn 2. Spring Chinese cabbage has stronger winterness compared with the autumn Chinese cabbage cultivars, and the elite allelic assembly of *BrVIN3.1* and *BrFLC1* was found to be a major genetic source of variation during selection [18]. In contrast to spring Chinese cabbage showing a very uniform genetic background, summer Chinese cabbage was more complex in its genetic composition; therefore, no clear genetic resource was identified [18]. However, in this study, resequencing data from Su et al. [18] was analyzed for the *BrFLC1* haplotype, in which summer Chinse cabbage showed a pure haplotype (H2: CA). The general characteristics of summer Chinese cabbage are more heat resistance and faster growth. In addition, summer Chinese cabbages are cultivated in Southern China where the temperature varies within a range of warm to high all year round; therefore, weak winterness ensures that they flower and produce seeds in this environment. Consistent with this growing habitat, the summer ecotype showed an opposite haplotype distribution from that of the spring ecotype. These results also indicated that the natural and artificial selection had posed very strong purification effects on the *BrFLC1* haplotype during the breeding of summer Chinese cabbage, and this might also be the case for caixin.

## 5. Conclusions

In summary, in this work, we demonstrated that *BrFLC1* haplotypes were related to *B. rapa* intraspecific diversification. Moreover, we showed that naturally occurred variations in *BrFLC1*, BrFLC1Pe1+58 (A/C), and BrFLC1Pi6+1 (G/A), both relate to flowering time, but each has an independent genetic effect. Our work will help to improve pre-mature bolting resistance in *B. rapa* vegetables.

**Supplementary Materials:** The following are available online at https://www.mdpi.com/article/10.3390/horticulturae7080247/s1, Table S1. SNPs and InDels in the BrFLC1 region detected in 199 *B. rapa* germplasm accessions. Table S2. DTB and DTF of the germplasm collection and the corresponding *BrFLC1*Pe1+58 genotype. Table S3. The genotype of *BrFLC1*Pe1+58 locus and *BrFLC1*Pi6+1 locus of 829 *B. rapa* germplasm accessions. Table S4. DTB and DTF of the $F_2$ population and the corresponding *BrFCL1* Pe1+58 allele genotype.

**Author Contributions:** Project designing, J.W. and X.W.; Data curation, J.L. (Jiahe Liu); Formal analysis, X.C. and J.W.; Investigation, J.L. (Jiahe Liu), Y.L., Y.C. and B.G.; Visualization, X.C.; Writing—original draft, J.L. (Jiahe Liu); Writing—review and editing, R.L., J.L. (Jianli Liang), X.W. and J.W. All authors have read and agreed to the published version of the manuscript.

**Funding:** This work was supported by the Agricultural Science and Technology Innovation Program (ASTIP), the Key Laboratory of Biology and Genetic Improvement of Horticultural Crops, Ministry of Agriculture, P.R. China.

**Institutional Review Board Statement:** Not applicable.

**Informed Consent Statement:** Not applicable.

**Acknowledgments:** This research work was supported the Science and Technology Innovation Program of the Chinese Academy of Agricultural Sciences, and the Key Laboratory of Biology and Genetic Improvement of Horticultural Crops, Ministry of Agriculture, P.R. China.

**Conflicts of Interest:** The authors declare no conflict of interest.

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
