# Peer review of "Selection on BrFLC1 Is Related to Intraspecific Diversity of Brassica rapa Vegetables"

_horticulturae, doi:10.3390/horticulturae7080247_

Round 1

Reviewer 1 Report

The manuscript describes interesting results as the development of markers for flowering is a major target for breeders and thus this could spead up the prossess. However the qustion here is how stricked is the regulation of flowering from the FLC locus and is it a qualitative characteristic or a quantittive.It is known from A. thliana that it is the levels of expression that control the flowering time rather the presence or absence of the transcript 

Furthermore it looks like for the graphs int he manuscript that both in early flowering and late flowering days there are both alleles suggesting that there is indeed a more complicated control mechanism

Author Response

Dear reviewer,

Thank you very much for assessing our manuscript “Selection on BrFLC1 is related to intraspecific diversity of Brassica rapa vegetables”. Regarding your valuable comments, our point-by-point responses are listed below:

Point 1: The manuscript describes interesting results as the development of markers for flowering is a major target for breeders and thus this could spead up the prossess. However the qustion here is how sticked is the regulation of flowering from the FLC locus and is it a qualitative characteristic or a quantittive. It is known from A. thliana that it is the levels of expression that control the flowering time rather the presence or absence of the transcript. 

Response 1: As far as we know, flowering time is a quantitative trait, but loss-of-function of FLC results effect in a qualitative pattern. For functional FLC, decreasing FLC expression level is the flowering regulation way in Arabidopsis thaliana. However, loss-of-function of FLC will definitely lead to early flowering in A. thaliana. The case is more complicated in B. rapa since there are four FLC copies. Multiple FLC copies showed a dosage effect. When one or two or three copies loss of function, the regulation node of FLC can still pose repressing effect, but different in repressing levels. This is one of the reasons that B. rapa shows wider flowering time variations than that of A. thaliana.

In this study, we identified a non-synonymous mutation at the first exon of BrFLC1 and demonstrated that it was related to early flowering in B. rapa. The previously identified splicing site mutation BrFLC1Pi6+1 leads to alternative splicing patterns and transcripts were quickly decayed. We are not clear about why this newly identified non-synonymous mutation leads to early flowering. The present study mainly focuses on identification of sequence variations that can be used for pre-mature bolting resistance breeding and the correlation between FLC loci and domestication of B. rapa.

Point 2: Furthermore, it looks like for the graphs in the manuscript that both in early flowering and late flowering days there are both alleles suggesting that there is indeed a more complicated control mechanism.

Response 2: We agree to the reviewer’s comment that there is indeed a more complicated control mechanisms of the two alleles for flowering time regulation. In the present manuscript, we observed both the two alleles are associated with flowering time variations, and we further confirmed that the BrFLC1Pe1+58 (A/C) locus poses independent genetic effects on flowering time. As the reviewer pointed out, it is more complicated to illustrate the mechanism of flowering time regulation in B. rapa.

Reviewer 2 Report

The authors showed that nonsynonymous mutation in BrFLC1 (BrFLC1Pe1+58) was related with flowering time in B. rapa. Further, the mutation seemed to tightly link with another mutation in BrFLC1 (BrFLC1Pi6+1). These mutations were considered to be important for enhancing premature bolting resistance in B. rapa. This manuscript gives important insights in the domestication of B. rapa through the selection of mutations in BrFLC. The manuscript is acceptable after minor revision.

  • BrFLC5 may be BrFLC3 in P.3 L.25.
  • Figure 2a seemed to be Figure 3a in P.10 L.8.
  • The color should be fixed to avoid the misreading of Figures. The genotype causing early flowering (BrFLC1Pe1+58(C)) should be blue in Fig. 2.
  • “2”should be subscript in “F2”.
  • “earlier” seemed to be “later” in P.13 L.9.
  • The conclusion of this manuscript is missing in last part of discussion. The author should state the importance of this report and the perspective of further research.

Author Response

Dear reviewer,

Thank you very much for assessing our manuscript “Selection on BrFLC1 is related to intraspecific diversity of Brassica rapa vegetables”. We have revised the manuscript according to your comments. Regarding your valuable comments, our point-by-point responses are listed below:

The authors showed that nonsynonymous mutation in BrFLC1 (BrFLC1Pe1+58) was related with flowering time in B. rapa. Further, the mutation seemed to tightly link with another mutation in BrFLC1 (BrFLC1Pi6+1). These mutations were considered to be important for enhancing premature bolting resistance in B. rapa. This manuscript gives important insights in the domestication of B. rapa through the selection of mutations in BrFLC. The manuscript is acceptable after minor revision.

Point 1: BrFLC5 may be BrFLC3 in P.3 L.25.

Response 1: We double checked the FLC gene located in non-collinear region between B. rapa and A. thaliana. It is BrFLC5 rather than BrFLC3.

Point 2: Figure 2a seemed to be Figure 3a in P.10 L.8.

Response 2: It has been corrected in the revised version.

Point 3: The color should be fixed to avoid the misreading of Figures. The genotype causing early flowering (BrFLC1Pe1+58(C)) should be blue in Fig. 2.

Response 3: We have fixed the color following the reviewer’s suggestion. The color has been changed into blue for   (BrFLC1Pe1+58(C)) in the revised version.

Point 4: “2” should be subscript in “F2”.

Response 4: We corrected all the 2 into subscript in F2 in the revised version.

Point 5: “earlier” seemed to be “later” in P.13 L.9.

Response 5: It has been corrected in the revised version.

Point 6: The conclusion of this manuscript is missing in last part of discussion. The author should state the importance of this report and the perspective of further research.

Response 6: A summary paragraph has been added at the end of the main text in the revised version.